# Recent Progress in the Discovery and Design of Antimicrobial Peptides Using Traditional Machine Learning and Deep Learning

**DOI:** 10.3390/antibiotics11101451

**Published:** 2022-10-21

**Authors:** Jielu Yan, Jianxiu Cai, Bob Zhang, Yapeng Wang, Derek F. Wong, Shirley W. I. Siu

**Affiliations:** 1PAMI Research Group, Department of Computer and Information Science, University of Macau, Taipa, Macau, China; 2Faculty of Applied Sciences, Macao Polytechnic University, Macau, China; 3Institute of Science and Environment, University of Saint Joseph, Estr. Marginal da Ilha Verde, Macau, China; 4NLP2CT Lab, Department of Computer and Information Science, University of Macau, Taipa, Macau, China; 5School of Pharmaceutical Sciences, Universiti Sains Malaysia, Pulau Pinang 11800, Malaysia

**Keywords:** antimicrobial peptide, machine learning, deep learning, classification, regression, therapeutic peptide, medicine

## Abstract

Antimicrobial resistance has become a critical global health problem due to the abuse of conventional antibiotics and the rise of multi-drug-resistant microbes. Antimicrobial peptides (AMPs) are a group of natural peptides that show promise as next-generation antibiotics due to their low toxicity to the host, broad spectrum of biological activity, including antibacterial, antifungal, antiviral, and anti-parasitic activities, and great therapeutic potential, such as anticancer, anti-inflammatory, etc. Most importantly, AMPs kill bacteria by damaging cell membranes using multiple mechanisms of action rather than targeting a single molecule or pathway, making it difficult for bacterial drug resistance to develop. However, experimental approaches used to discover and design new AMPs are very expensive and time-consuming. In recent years, there has been considerable interest in using in silico methods, including traditional machine learning (ML) and deep learning (DL) approaches, to drug discovery. While there are a few papers summarizing computational AMP prediction methods, none of them focused on DL methods. In this review, we aim to survey the latest AMP prediction methods achieved by DL approaches. First, the biology background of AMP is introduced, then various feature encoding methods used to represent the features of peptide sequences are presented. We explain the most popular DL techniques and highlight the recent works based on them to classify AMPs and design novel peptide sequences. Finally, we discuss the limitations and challenges of AMP prediction.

## 1. Introduction

Antimicrobial peptides (AMPs), also known as host defense peptides, are a diverse class of naturally occurring molecules discovered in animals, plants, insects, and even in microorganisms [1,2]. They protect the host from a broad spectrum of microbial pathogens by directly killing them or indirectly modulating the host’s defense systems. Due to their natural antimicrobial functions and low probability for drug resistance [3], AMPs are considered promising alternatives to antibiotics. Some AMPs also exhibit cytotoxicity towards cancer cells, which suggests that they are potential sources of therapeutic agents for cancers [4]. To date, more than 3400 AMPs have been identified from natural sources and cataloged in the Antimicrobial Peptide Database (APD) [5]. The total number of validated AMPs recorded in public databases (e.g., dbAMP and DBAASP), including artificial AMPs tested in synthetic chemistry studies, has exceeded 18,000 entries [6,7]. However, the number of natural AMPs is probably in the order of millions [8], so new strategies are needed to facilitate the discovery and design of novel AMPs.

Computational methods are playing an increasingly important role in AMP research. In particular, traditional ML and DL methods are considered efficient methods for recognizing previously unknown patterns from sequences, which help to predict the antimicrobial potential of new sequences. In this paper, traditional ML refers to all ML methods such as support vector machine (SVM), k-nearest neighbor (kNN), random forest (RF), and single-layer neural network (NN), but not DL methods, which usually contain multiple layers of NN. Based on the prediction results, candidate sequences can be selected and prioritized for experimental validation, greatly reducing the time and cost in the search for new active AMPs. The main advantage of a computational study is that investigation is not limited to known peptides. While mutations or modifications based on template sequences can be performed to optimize antimicrobial potency, random libraries with a virtually unlimited number of sequences can also be explored in search of new AMP motifs.

In this review, we make a brief introduction to traditional ML methods and discuss the recent advances in the development of DL methods for discovering and designing AMP sequences. We focus on the methodological aspects of the proposed methods and highlight works associated with experimental investigations that ultimately contributed to the identification of novel AMPs. For better understanding, we first introduce AMPs, their discovery, classification, mechanism of action, therapeutic applications, and limitations. For a more comprehensive overview of the biology of AMPs, see references [1,8,9].

### 1.1. Discovery of Early AMPs

The presence of antimicrobial substances in nature was first recognized by Alexander Fleming in 1922 [10] when he found that his nasal mucus could prohibit bacterial growth. The substance was an antimicrobial protein, named lysozyme, which has the ability to rapidly lyse bacteria without being toxic to human cells. It can be found in tissues and physiological fluids of animals and in egg whites, which were later confirmed to play an important role in the innate immune system [11]. Soon after Fleming’s discovery, in 1928, the first bacterium-produced AMP nisin was identified by Rogers and Whittier in fermented milk cultures [12]. It is a cationic peptide produced by *Streptococcus lactis* that exhibited potent bactericidal effects against a wide range of Gram-positive foodborne bacteria, thus making it an important food biopreservative [13]. Nisin has also been found to have important biomedical applications, including bactericidal activity against the superbug methicillin-resistant *Staphylocococcus aureus* (MRSA) [14]. In 1939, René Dubos reported another AMP gramicidin, which was isolated from the soil bacterium (Gram-positive) *Bacillus brevis* [15]. Gramicidin was a heterogeneous mixture of six AMPs consisting of N-formylated polypeptides with alternating L- and D-amino acids. It exhibited both bactericidal and bacteriostatic activities against a wide range of Gram-positive bacteria and was the first AMPs to be commercially produced as antibiotics despite its high cytotoxicity [9].

The discovery of AMP was not limited to bacteria. Kiss and Michl discovered the first animal AMP brombinins in the 1960s from the venomous skin secretion of the orange-speckled frog *Bombina variegate* 1960s [16]. Zeya and Spitznagel detected a class of AMPs in the neutrophils of a rabbit and a guinea pig in the 1960s [17], and the same family of AMPs (HNP-1, HNP-2, HNP-3) was later identified by Ganz and Lehrer from human neutrophils in the 1980s, which were named defensins [18]. The Boman group identified cecropins from the hemolymph of pupae of the giant silk moth by injecting *Enterobacter cloacae* [19]; which was the first report of an α-helical AMP, and Zasloff found magainins that were secreted in the skin of the African clawed frog *Xenopus laevis* [20]. Venom is a tool of self-defense for venomous animals, such as spiders, scorpions, and bees. The first spider AMPs were identified in the wolf spider *Lycosa carolinensis* in 1998 and were named lycotoxins [21]. They showed the ability to kill both bacteria (*Escherichia coli*) and yeasts (*Candida glabrata)* at micromolar concentrations by forming pores in membranes. In addition to defense against infectious microbes, they caused the efflux of calcium ions from rat brain synaptosomes, suggesting that they might contribute to paralysis of envenomated prey [21].

The production of AMPs is also a defense strategy of plants. They are found in the roots, seeds, flowers, stems, and leaves of a wide variety of plants and have bactericidal activity against phytopathogens [22]. The first plant AMP, called purothionin, was isolated from wheat flour by Balls et al. in 1942 [22,23]. Later in 1990, another class of AMPs isolated from wheat endosperm by Collila and Mendez [24], originally called gamma-purothionins, was found to share high structural properties with mammalian and insect defensins (and were renamed plant defensins) [25].

### 1.2. Classification

There are numerous ways to classify AMPs based on their origins, sequence properties, structural properties, biological activities, and molecular targets. APD3 is a manually curated database of AMPs of natural origin with experimentally validated activity (MIC < 100 μM) [5]. AMPs can be classified in various ways. According to APD3 (see https://aps.unmc.edu/classification (accessed on 1 May 2022)), the seven types of classification are: (1) biosynthetic machines (gene-coded or non-gene-coded), (2) source organisms (bacterial, plant or animal), (3) biological functions (antibacterial, antiviral, antifungal, anti-parasitic, etc.), (4) peptide properties (amino acid composition, length, hydrophobicity, charge, and length), (5) covalent bonding patterns (linear peptides, sidechain-sidechain linked, sidechain-backbone linked, circular peptides), (6) secondary structures (α, β, αβ, and non-αβ), and (7) molecular targets (cell surface-targeting and intracellular targeting) [26].

Since the structural organization of AMPs plays an important role in the mechanism of action for molecular function [27], we present this classification in detail below. Based on the two basic secondary structures, an AMP can be categorized into four classes depending on whether it contains α-helix, β-sheet, both α and β (i.e., mixed), or no α and β (e.g., coil) [26]. According to the records of natural peptides in the APD database (see Table 1), the percentages of α, β, αβ, and non-αβ peptides among the known AMP 3D structures are 69%, 12%, 16%, and 3% respectively. The class with the greatest average length is αβ (59 aa), followed by β (35 aa); α and non-αβ peptides have average lengths of 30 aa and 27 aa, respectively. Most known AMPs are cationic, with a minority of peptides carrying neutral or anionic charges. The α and β peptides have an average net charge of +3.60, αβ peptides have a higher average net charge of +5.40, while non-αβ peptides have +2.55. As shown in Figure 1, α peptides are rich in lysine, leucine, and alanine, whereas β peptides are rich in cysteine and arginine. It is noteworthy that all four classes have high proportions of glycine, suggesting that it is important for both structural support and flexibility of peptides for antimicrobial functions.

#### 1.2.1. α-Helical Peptides

The α-helical peptides represent the largest class of AMPs and are also the most-studied one. Many α-helical AMPs are linear and amphipathic, consisting of cationic and hydrophobic amino acids spatially segregated on the opposite faces of the helix [28]. Prominent examples of this class are the frog magainins [20], the mammalian cathelicidins [29], the moth cecropins [30], and the bee melittin [31]. These AMPs exhibit a strong affinity for membranes, thereby compromising the stability of the bilayer, disrupting membrane organization, and/or forming pores [32]. Although defined as the α-helix class, these AMPs may not be α-helix at the inactive state. Magainin 2, for example, is initially disordered in an aqueous solution but folded upon binding to a membrane [33]. The orientation of the helix at the membrane is concentration-dependent. PGLa, a member of the magainin family, first lies parallel to the membrane surface, whereas, at high concentrations, it rotates about the membrane, to insert into the membrane at a certain tilt angle [34]. In contrast, the human cathelicidin LL-37 can adopt a partially helical structure in the solution, hence, obliging it to oligomerize with other peptides to hide the hydrophobic surface [29]. Apart from individual activity, helical peptides of different sequences can act synergistically resulting in enhanced cytolytic and antibacterial effects [33].

#### 1.2.2. β-Sheet Peptides

This group of peptides includes at least two β-strands forming a β-sheet conformation. The structure is stabilized by one or more disulfide bonds formed by pairs of cysteine amino acids arranged side-by-side on the neighboring strands. Similar to helical AMPs, the hydrophobic and polar residues are arranged in clusters on spatially separated surfaces of the peptide. Depending on the structural characteristics, they are further divided into subgroups: β-hairpin and α-, β-, and cyclic θ-defensins. β-hairpin AMPs adopt the common characteristic of anti-parallel β-sheets linked by a small turn of three to seven amino acids forming a hairpin shape [35]. There are AMPs with single disulfide bonds (e.g., lactoferricins, bactenecin, tigerinin, arenicins, thanatin), two disulfide bonds (e.g., arenicin-3, tachyplesins, polyphemusins, gomesin, androctonin, protegrins), three and four disulfide bonds (e.g., hepcidins). The key role of the disulfide bonds is to provide structural stability and peptide resistance to biodegradation [36].

Defensins are important members of this class and the next AMP class (αβ mixed). α-, β-, and θ-defensins all contain largely β-sheet structures and three disulfide bonds that differ in connectivity between cysteine residues. In particular, θ-defensins are cyclic peptides, similar to the joining of two β-hairpins [37]. As the two β-strands in θ-defensins are highly constrained, the cyclic backbone (called the cyclic cystine ladder) is very rigid, and might have a role in molecular recognition and antibacterial activity [37,38]. Although the antibacterial activities of θ-defensins are comparable to those of other AMPs, the symmetric cyclic scaffold with superior stability offers an opportunity to design peptide drugs with bioactive epitopes for activity and specificity [39,40].

#### 1.2.3. αβ Mixed Peptides

This class contains peptides with mixed α-helix and β-sheet structures. For example, the human β-defensin-3 contains three β-strands and a short helix in the N-terminal region [41]. Defensins from plants and some invertebrates exhibit a conserved structural cysteine-stabilized αβ motif (CSαβ) [42], which is composed of an α-helix followed by two anti-parallel β-strands and is stabilized by three or four disulfide bridges [43]. Interestingly, these CSαβ-containing defensins from plants are predominantly active against fungi, whereas those from insects are predominantly active against bacteria [43]. Although the role of the CSαβ-motif is unclear, AMPs with the motif have been observed to act with a common mechanism of action by inhibiting cell-wall formation and binding to Lipid II [43].

#### 1.2.4. Non-αβ Peptides

This class of peptides do not adopt well-defined secondary structures. They are rich in tryptophan, glycine, proline, threonine, serine, and histidine amino acids, and exhibit high flexibility in aqueous solution (see Figure 1). Tryptophan residues are known to have a strong preference for the interfacial region of lipid bilayers. They play an important role in membrane penetration by associating with the positively charged choline headgroups of the lipid bilayer and forming hydrogen bonds with both water and lipid bilayer components when in the interfacial region [44]. A well-known peptide of the non-αβ class is indolicidin isolated from bovine neutrophils, which is rich in both Trp (39%) and Pro (23%) residues. It does not adopt canonical secondary structures but presents unique, extended, membrane-associated peptide structures. In the large unilamellar phospholipid vesicles (DPC), the backbone of indolicidin forms a wedge shape with the Trp and Pro residues clustered to form a central hydrophobic core, bracketed by positively charged regions near the peptide termini [45]. Indolicidin was proposed to penetrate bacterial membranes and bind to the negatively charged phosphate backbone of DNA, thereby inhibiting DNA synthesis and inducing filamentation of bacteria [46].

### 1.3. Mechanism of Action

AMPs have attracted attention as potential antimicrobial agents as they kill bacteria with a different mechanism of action (MOA) than conventional antibiotics. The MOA can be divided into three types: membrane disruption, metabolic process interference [47], and immunomodulation [48].

AMPs are selective for bacterial membranes primarily through electrostatic interactions. In contrast to mammalian cell membranes, bacterial cell membranes contain abundant negatively charged components such as phosphatidylserine (PS), phosphatidylglycerol (PG), cardiolipin (CL), and teichoic acid (TA) in the peptidoglycan cell wall of Gram-positive bacteria, and the endotoxin lipopolysaccharide (LPS) (in the outer membrane of Gram-negative bacteria) [8]. The strong electrostatic interaction between the cationic peptides and the anionic surface of bacterial membranes facilitates initial peptide binding [47]. Subsequently, AMPs exert membrane interruption via three models of perturbation: barrel-starve, toroidal, or carpet models [1,8]. In the barrel-stave model, AMPs form a bundle, which is inserted into the membrane to form a hydrophilic pore, with the hydrophobic residues interacting with lipids and the polar residues facing the pore channel. The toroidal model forms pores by inducing thinning and curvature in the membrane with the lipid headgroups bent towards the membrane core so that the pore is lined by both the peptides and the lipid headgroups. The carpet model, as the name suggested, covers the membrane surface without penetrating. It causes tension on the membrane leading to membrane disintegration and micelle formation.

Instead of membrane disruption, some AMPs translocate across the bacterial membrane and bind to intracellular targets that affect specific enzymatic activities or vital metabolic processes, such as the synthesis of DNA, RNA, proteins, and cell walls [47]. Other mechanisms of membrane disruption have also been reported, such as the formation of aggregates, electroporation, and alteration of the distribution of membrane components [8,49].

### 1.4. Therapeutic and Industrial Applications

#### 1.4.1. Biomedicines in Pharmaceutical Industry

AMPs are considered promising alternatives to traditional antibiotics given their potency, broad-spectrum activity, multiple modes of action, and low chance of resistance development. The first AMP drug, Gramicidin A, isolated from the soil bacteria *Bacillus subtilis*, was manufactured commercially in the 1940s [50]. It is still used today for topical treatment of superficial wounds and infections of the eyes, nose, and throat. Due to its high hemolytic activity, it cannot be administered internally as a systemic antibiotic [51]. Nisin from *Lactococcus lactis* was first commercially marketed in 1953 as an antimicrobial agent but later found its use as a safe food biopreservative (see below) [52]. Because of good safety, the use of nisin A has been extended to non-food bacteria in the context of infectious diseases, including drug-resistant bacteria strains, cancer, and oral caries [13]. An increasing number of bioengineered variants of nisin are reported to have promising therapeutic potential for infectious diseases associated with *S. pneumoniae, enterococci, C. difficile,* etc., and may work synergistically with antibiotics, such as ciprofloxacin and vancomycin [13]. Other approved AMPs, such as antimicrobials include polymyxin B (in 1955), polymyxin E (better known as colistin, in 1962), and daptomycin (in 2003) [53].

#### 1.4.2. Substitutes for Antibiotics and Pesticides in Agriculture
and Animal Husbandry

The excessive use of antibiotics in the agriculture and animal production industry in recent years has raised serious concerns over the emergence of antibiotic-resistant bacteria and the increase of health and environmental risks. AMPs are considered as substitutes for conventional antibiotics, to be used as veterinary and plant medicines. Many studies have focused on naturally produced AMPs from animals and plants, which have sustained resistance to pests and pathogens and are safe for host organisms without having environmental side effects [54]. There is growing evidence that genetically modified animals and plants with AMP-expressing genes (i.e., transgenic) confer the organism’s resistance to microbial pathogens. For example, Peng et al. [55] demonstrated that the recombinant porcine β-defensin 2, when used as a medicated feed additive, improved growth and intestinal morphology of weaned piglets, and reduced post-weaning diarrhea in piglets. Transgenic pigs over-expressing this AMP had improved resilience to *Glaesserella parasuis* infection, with alleviated lung and brain lesions and reduced bacterial loads in the lung and brain tissues [56].

#### 1.4.3. Food Preservatives and Packaging in the Food Industry

Antimicrobial agents are essential ingredients for the preservation of foodstuff in the modern food industry. Nisin is widely used in dairy and meat products to control contamination from *Listeria* strains [52]. It is still the only bacteriocin legally approved as biopreservative (E234) and has been approved as GRAS (Generally Recognized As Safe) by the US Food and Drug Administration (FDA) in 1988 [52]. In addition to their functions as preservatives, AMPs have also gained attention as potential ingredients in food packaging. Using active packaging techniques, AMPs can be incorporated into the encapsulation systems (e.g., nanocarriers, emulsions, films) and released in a slow, controlled manner to inhibit foodborne pathogens, thereby extending the shelf life of food [57].

### 1.5. Limitation of AMPs and Bacterial Resistance

While AMPs have been proposed as a promising alternative for bacterial therapeutics, there are certain limitations with AMPs that hinder their success in the development into drugs. Major limitations include high production costs, low stability due to proteolytic degradation, cell toxicity, and susceptibility to physiological conditions of the host, such as pH and ion concentration.

Even if the probability is low, bacteria may still evolve to recognize and respond to the bactericidal effects of AMPs. A notable mechanism in Gram-negative bacteria to detect the presence of cationic AMPs and other environmental signals is the PhoP/PhoQ two-component regulatory system (TCS) [58]. PhoQ responds to these signals by autophosphorylation and activates PhoP to regulate the expression of downstream outer membrane protein and lipid contents in the bacterial envelope, thereby controlling membrane polarity and stiffness to resist invasion by AMPs. Other mechanisms include the use of capsular polysaccharides and other external molecular structures that act as protective shields, the use of transporters that pump AMPs out of the cell, proteolysis of AMPs, and suppression of their expression in host cells. Similar arsenals have been exploited by Gram-positive bacteria to overcome AMP activity as thoroughly reviewed [59].

## 2. AMP Discovery and Design—The Machine Learning Workflow

In recent years, the search for known or predicted peptide sequences with the desired properties has become very popular, and the corresponding approaches are constantly being developed. Here, advanced computational strategies are presented and two groups of research approaches are distinguished: first, the discovery of new AMPs from naturally occurring sequences; second, the design of artificial AMPs by modification of known AMPs or design de novo. The AMP discovery approach predicts potential peptides by virtually screening large libraries of known peptides, specifically looking for peptides that are structurally closest in sequence to known AMPs. The AMP design approach generates artificial AMPs in an evolutionary manner.

Both research approaches follow the logical flow of a pipeline shown in Figure 2, which starts with encoding the inputs, constructing the traditional ML or DL model, predicting the biological activity, or generating new peptides. First, we present the common features of these two approaches, i.e., input coding. Then we outline the methods used in various ML models for these two approaches. These approaches have addressed difficult problems related to primary sequence spaces and peptide structures while economically delivering AMPs with broad spectrum activity.

## 3. Feature Encoding Methods

The most fundamental data of AMPs are the sequences; their derived data include sequence compositions [53,60], physicochemical properties [61,62,63], etc. Given the importance of information sources for computational prediction, the representation of biological or chemical data for use in the discovery and design of AMP is an essential component in the flow of a ML pipeline. Feature encoding methods generate numerical features from peptide sequences to prepare them for ML. According to Singh et al. [64], feature encoding methods fall into two broad categories: peptide-level features and amino acid-level features.

### 3.1. Peptide-Level Features

Peptide-level features can be further classified into sequence-based and structure-based features.

#### 3.1.1. Sequence-Based Features

The sequence-based features compute feature vectors based on the compositions of amino acids or amino acid groups. These kinds of feature-encoding methods include one-hot encoding [65], general and pseudo amino acid compositions [66], reduced amino acid compositions [67], etc. Among a variety of input encoding methods, one-hot encoding is one of the most popular methods that retain the information on the order of amino acids [68,69,70]. Each amino acid is represented as a 20-bit binary vector. For example, alanine (A) is represented as the vector [10000000000000000000] where every entry is “0” except for a “1” at the index of the amino acid of interest. General amino acid compositions encode the frequencies of 20 natural amino acids. Reduced amino acid compositions [66] consider not only 20 amino acid compositions but also a set of discrete sequence correlation factors. Pseudo-reduced amino acid compositions compute the occurrence number of each clustered similar amino acid group [67]. Hybrid approaches that used one-hot encoding together with physicochemical features were also successful in the prediction of antimicrobial peptide activity [71].

#### 3.1.2. Structure-Based Features

The structure-based features further consider the structural features of the peptide residues. These kinds of feature encoding methods include protein secondary structures [72], quantitative structure–activity relationship (QSAR) [73], distance distribution [74], and so on. Protein secondary structures record each amino acid as α-helix, β-sheet, or random coil [72]. QSAR reveals the relationships between chemical structures and biological activities [73]. Distance distribution describes the distribution of distances between each type of atom [74]. Moreover, simplified molecular input-line entry system (SMILES) codes [75,76] are well-known chemical codes used to annotate compound structures. This representation encodes structures of chemical species using a simple text string, and several databases have provided peptide sequences in the SMILES format for immediate use [77]. Recently, some comparative studies have been conducted to summarize and compare commonly used peptide encoding methods for peptide classification [78,79].

### 3.2. Amino Acid-Level Features

The amino acid-level feature of a peptide is the sequence itself. Each word is the one-letter code of an amino acid, similar to words in a sentence. These features are generally used by sequence-based DL algorithms, such as recurrent neural networks (RNNs) to build a classification model by analyzing sequence data [64]. Apart from directly using different RNN layers after inputting peptide sequences, many works also use embedding layers to extract representative features, such as word embedding (e.g., Word2vec, Bag-of-Words (BoW) [80]), and contextualized embedding (e.g., BERT). These natural language processing (NLP) techniques create one-dimensional vectors for every word (i.e., an amino acid) in sequences without prior knowledge of biology. This results in a more compact representation of the input, where semantically similar words are placed close to each other in the vector space, improving both accuracy and efficiency in learning.

#### 3.2.1. Word Embedding

Word2vec is one of the most popular methods for generating word embeddings. The goal is to capture the contextual meaning of the words by creating a lookup table, called the word embedding matrix, which consists of a list of words and their corresponding learned representation of the word. There are two ways to generate the learned representations using shallow neural networks [81]: the continuous Bag-of-Words (CBOW) algorithm and the skip-gram algorithm. The CBOW model learns the embedding by predicting the current word (e.g., an amino acid in the middle of a sliding window) based on its context (amino acids in the sliding window, except the middle one) whereas the skip-gram model learns by predicting the surrounding words given a current word. The optimized weights in the network are the learned representations that are then used in downstream ML tasks.

For example, Veltri et al. [80] used the CBOW method to assign a unique numerical token to each amino acid in a peptide sequence. Sharma et al. [82] proposed Deep-ABPpred using the skip-gram model to create word embeddings and bidirectional long short-term memory (Bi-LSTM) to predict AMPs.

#### 3.2.2. Contextual Embedding

Contextual embedding methods move beyond word-level semantics in that each learned embedding is a function of the entire input sequence. They capture word uses across various contexts and yield different representations of the same word in different contexts [83]. Popular DL architectures for generating contextual embeddings include sequence-to-sequence, LSTM, and transformer [84]. Often a large unlabeled dataset is used to pre-train the model, then the model is transferred by tuning the model to the specific prediction task at hand, for which there are often less data available [85]. Or the generated representations are used as features for task-specific architectures [83].

In developing models for AMP prediction, Zhang et al. [86] obtained 556,603 protein sequences from UniProt [87] as pretraining samples. They generated *k*-mer as words with k=1,2,3. BERT [88] was used to train a deep bidirectional language representation model for two tasks, masked language model (MLM) and next sentence prediction (NSP), to capture word-level and sentence-level representations, respectively. Finally, the output layer of the pre-trained model was changed and fine-tuned to suit downstream prediction tasks.

Along the same line of research, Dee et al. [89] used language representation models trained with the UniRef100 and UniRef50 protein databases, which consist of 216 and 45 million protein sequences, respectively. More language model pretraining techniques were tried, including bidirectional encoder representations from transformers (BERT) [88], text-to-text transfer transformer (T5) [90], and the auto-regressive model (XLNet) [91], and the convolutional neural network (CNN) was used as the classifier. The authors found that T5 trained on UniRef50 generated the highest accuracy, suggesting that using the whole transformer architecture to build the pre-trained language model was better than the encoder-only (BERT) or decoder-only (XLNet) models [89].

## 4. AMP Prediction by Traditional Machine Learning

AMP discovery from large-scale natural known peptide libraries is based on the antimicrobial activity prediction from traditional ML models in a screening manner. Traditional ML techniques, such as SVM [92,93,94,95,96], discriminant analysis (DA) [97], RF [98,99,100,101], kNN [95,102,103], and ensemble learning [104,105,106,107,108] have been applied to discover AMPs by classification. Among these methods, SVM non-linearly transforms the original input space into a higher-dimensional feature space by means of kernel functions [109,110]. With its powerful performance in handling noise, it has been increasingly used for the classification of biological data [92,97]. Unlike SVM, which uses a nonlinear transformation, DA uses a linear combination of independent variables to predict group membership for categorical dependent variables (i.e., class labels) [111,112]. RF is a combination of decision trees, and each tree is generated with sub-samples of the dataset [111]. While kNN is an instance-based learning method, it stores all available cases and classifies an unknown example with the most common class among k closed examples, and the selection of k and the distance function is crucial [112].

For a better understanding of the performance of various traditional ML models, there is research that uses different traditional ML models simultaneously. Early, Thomas et al. [97] created a large AMP dataset containing both sequences and structures of AMPs. Their comparative study showed that SVM, RF, and DA achieved test accuracy of 91.5% (SVM), 93.2% (RF), and 87.5% (DA), respectively. Recently, Kavousi et al. [95] developed the IAMPE platform to predict AMPs. This platform employed Naïve Bayes (NB), kNN, SVM, RF, and XGBoost to build a classification system fed by peptide features, including composition and physicochemical properties. The highest prediction accuracy of the combined features on the benchmark dataset achieved a very high accuracy of 95%. Meanwhile, Xu et al. [100] also presented a comprehensive evaluation of traditional ML-based methods with five-fold cross-validation (CV) results showed that RF, SVM, and eXtreme gradient boosting performed better in learning AMP sequences.

## 5. AMP Prediction by Deep Learning

Unlike traditional ML techniques that require prior domain knowledge and well-engineered input features, deep neural networks can automatically learn high-level features and have been used in many bioinformatics tasks [113]. The early study by Fjell et al. [114] used QSAR descriptors and fed them to an artificial neural network that predicted peptide activity against *P. aeruginosa*, and achieved 94% accuracy in identifying highly active peptides. Since then, many DL methods have been proposed for predicting AMps [80,89,100,114,115,116,117,118,119].

To illustrate the AMP discovery with DL methods in detail, we discuss deep neural networks (DNNs), DL with CNN layers, DL with RNN layers, hybrid learning, and other DL approaches for identifying AMPs. Since most of the research papers addressed the AMP classification problem, but only a few of them investigated regression for predicting different biological activity values of AMPs, the DL for AMP regression is discussed in a separate subsection. Deep models do not always outperform the so-called shallow models, such as SVM and RF in the classifications of AMPs [120], and it has been suggested that DL should be used only when significantly better performances have been demonstrated when computational costs are taken into account.

### 5.1. Deep Neural Networks (DNNs)

In this work, we refer to DL architectures with only dense layers (also as fully connected layers), i.e., the neural network, as DNNs. A DNN requires multiple learning layers to train a complex and non-linear function [121]. It consists of an input layer, multiple hidden layers, and an output layer. Each layer has a set of neurons that perform processing. The input neurons take numerical values representing various features of the data and pass the information to the first hidden layer. Each neuron in the hidden layer and the output layer processes the collected information using a weight vector and a bias vector. The generation of the output is based on an activation function similar to that of the neurons in our brain, so that a signal is generated only when the accumulated value exceeds a certain threshold. The strength of the learning comes from the different weight and bias vectors of all the neurons, which can focus on different patterns in the data. Combining the results of these neurons in the output layer produces a prediction that is compared to the ground truth. Errors in the prediction are propagated back from the output layer to the hidden layers to adjust the vectors and minimize the errors in a number of learning cycles until the network converges.

Several recent works developed AMP prediction methods using DNN architectures. Timmons et al. [122] proposed a DL method with eight different neural dense layers, called ENNAACT, with physicochemical features [123] as input for identifying anticancer peptides (ACPs). ENNAACT showed the highest performance with 98.3% accuracy at 10-fold cross-validation based on the ENNAACT dataset compared to RF, SVM with linear kernel and SVM with RBF kernel. In addition, Timmons et al. [124] presented ENNAVIA, a DL method that uses three dense layers with AAC, DPC, AAindex [125], and physicochemical properties [123] as input features, for predicting the activity of anti-virus peptides (AVPs). ENNAVIA achieved the best performance compared to the other state-of-the-art methods with 95.7% accuracy in a validation dataset consisting of 60 positive and 60 negative sequences compared to the other state-of-the-art methods. Ahamd et al. [121] proposed a DL method with three dense layers (called Deep-AntiFP) for predicting anti-fungal peptides (AFPs); its input features were generated by three different feature encoding methods: composite physicochemical properties (CPP) [126], quasi-sequence order (QSO) [127], and reduced amino acid alphabet (RAAA) [128]. Furthermore, Deep-AntiFP achieved 94.23%, 91.02%, and 89.08% accuracy based on the training, alternate, and independent datasets, respectively. The proposed Deep-AntiFP outperformed the other existing models and achieved the highest performance.

### 5.2. Deep Learning with CNN Layers

DL with CNN layers [129,130,131] has proven useful in predicting AMP. CNN is able to handle high-dimensional data with convolutional kernels. It can reduce data dimensions and extract local information well, but it ignores long-term dependencies in the data [132]. Many works using DL methods employed a varying number of CNN layers and dense layers as the base architecture. A convolutional layer aims to learn a feature representation of the inputs using filters. Each filter (also called a convolution kernel) systematically convolves with the input field over the entire input matrix to produce a feature map. The full feature maps are obtained by using several different filters to extract different features from the inputs. After a convolutional layer, a pooling layer (e.g., average pooling or max pooling) is added to reduce the resolution of the generated feature map. By stacking multiple convolutional layers and pooling layers, higher-resolution feature representations can eventually be extracted. As in DNN, the dense layers aim to perform reasoning and provide information to the output layer to produce the final prediction result.

Both encoding methods and embedding methods can be used to represent sequences numerically. Often a systematic investigation [133] is performed to analyze the significance and contribution of an encoding method or a layer to finally confirm the resulting architecture. If more than one encoding method proves informative, these features can be concatenated, either in the first layer (i.e., the input layer), after the CNN layers, or even after the output layer where the output results are combined to produce the final prediction. Similar to DNN, the training of CNN is done via global optimization of the network parameters by minimizing a selected loss function.

In our previous work on AMP prediction, we developed a CNN model, called Deep-AmPEP30 [115], for short-length peptides (≤30 amino acids) prediction. It was designed with two convolutional layers, each followed by a max pooling layer. The max pooling layer can help select features that have the highest value, i.e., are most informative. The convolution results were flattened and fed into the dense layer to output a probability as the result. The pseudo-K-tuple reduced amino acid composition (PseKRAAC) was selected as the encoding method after a comprehensive feature comparison. Using Deep-AmPEP30 as the engine for genome screening, we successfully identified a potent AMP with 20 residues from the genome sequence of *Candida glabrata*.

Su et al. [134] proposed a deep CNN network with an embedding layer for encoding sequences, the multi-scale convolutional layers for capturing sequence patterns, followed by standard pooling layers and a dense layer. The embedding layer converted each amino acid into a numeric vector of real numbers (as opposed to one-hot encoding, which only includes 0 and 1 s). This dense representation captured semantic information about amino acids and relationships between different amino acids [134]. The multiple convolutional layers used varying filter lengths to ensure that motifs with different lengths could be learned. The proposed model was found to outperform state-of-the-art models with 92.2% accuracy on the APD3 benchmark dataset [5]. In addition to the CNN model, the authors proposed a fusion model that generated predictions based on the concatenation of the results of the CNN part and the DNN part, the latter using the conventional AAC and DPC features. However, the fusion model showed only a small improvement (<1%) in model accuracy.

Dua et al. [133] proposed the deep CNN method with one-hot encoding to generate the input features for AMP identification. Interestingly, by systematic comparison, they showed that CNN performed better than different variants of RNN models, including simple RNN, long short-term memory (LSTM), LSTM with a gated recurrent unit (GRU), and bidirectional LSTM (Bi-LSTM) over Veltri’s test dataset [80].

Regarding works on specific activity predictions for AMP, such as anticancer, Cao et al. [135] proposed DLFF-ACP, a DL method using the DNN and CNN for predicting probabilities of ACPs. DLFF-ACP contains two input channels (also called the branch). The handcrafted feature channel accepted a selection of amino acid compositional features, including AAC, DPC, and CKSAAGP. The CNN channel encoded each amino acid in a sequence into a number of 1 to 20 and processed by an embedding layer and convolutional layers. Predictions from the two channels were combined and further learned to classify ACP. DLFF-ACP achieved an accuracy of 82% with a 10-fold CV on its training dataset and performed on par with the state-of-the-art methods on its test dataset.

Lin et al. [119] proposed AI4AMP trained on sequences encoded with a combined physicochemical property matrix called PC6. The properties, namely hydrophobicity, volume of side chains, polarity, pH at the isoelectric point, pKa, and the net charge index of side chains, were selected from six clusters of properties based on the result of hierarchical clustering. The PC6 encoding method combined with a DL architecture consisted of a convolutional layer, a LSTM layer, and a dense layer. The proposed model showed a competitive performance compared to the word embedding-based model (word2vec). PC6 with CNN was also applied to ACP prediction and resulted in the AI4ACP method [136], which showed a stable performance and high accuracy.

For AVP predictions, CNN also performed well. Sharma et al. [137] proposed Deep-AVPpred, a DL classifier for discovering novel AVPs in peptide sequences. Deep-AVPpred used the concept of transfer learning with a one-dimensional CNN architecture. To learn different relationships between amino acids, the authors utilized multiple kernels of different sizes and a set of 200 filters for each kernel size. Sequence encodings were achieved using pretrained embeddings constructed from unsupervised learning of millions of UniRef50 protein sequences [138].

Instead of individual classifiers for different functional activities, a deep neural network was proposed that served to identify up to 20 bioactivity classes of peptides, including antimicrobial (antibacterial, antifungal, antiviral, anti-parasitic), biological (ACE inhibitor, antifreeze, antioxidant, hemolytic, neuropeptides, toxic), and therapeutic (anticancer, anti-hypertensive) activities. The method, named MultiPep [139], was constructed using the dendrogram template obtained by hierarchical clustering of the collected peptide activity data and each cluster was ’learned’ by a class-clade-specific CNN.

### 5.3. Deep Learning with RNN Layers

Since peptide sequences are chains of amino acid letters similar to human language, it is natural to apply successful language processing techniques to sequence processing and prediction tasks. Among DL architectures, RNN and its variants (LSTM, Bi-LSTM, GRU, etc.) are specifically designed for variable-length sequence processing. The core idea of RNN is the use of a cyclic connection so that the current state of the RNN cell can be updated based on previous states and new input data. This feedback feature allows RNN to capture positional dependencies between information in a data sequence. For AMPs, RNNs can learn remote dependencies inside sequences, but they suffer from vanishing gradients [140]. Nevertheless, RNN has the pitfall that when input data are separated by large temporal gaps, their relationship cannot be connected. To solve the short-term memory problem, LSTM [141] was proposed, which consists of an input gate, a forget gate, and an output gate. These gates allow the LSTM cell to selectively discard information in the cell state that was learned in previous timesteps and decide what new information to add to the cell state or carry forward to the next time step as a result. As the name implies, a bidirectional LSTM (BiLSTM) can process sequence data in both forward and backward directions by using two LSTMs and an additional layer that concatenates outputs from both LSTMs. BiLSTM is superior to LSTM due to the additional layer and dual exposure of data for training, suggesting that BiLSTM is able to capture information that may be missed otherwise in unidirectional training. A gated recurrent unit (GRU) is a somewhat simplified version of LSTM where only two gates, update and reset gates, are present.

Many works successfully applied RNN to improve AMP predictions. Sharma et al. [82] proposed Deep-ABPpred using BiLSTM with word embedding (word2vec) to identify AMPs after a comprehensive comparative study. A number of BiLSTM models coupled with amino acid level features (word2vec, one-hot encoding, PAM250, and BLOSUM62), and SVM/RF models with peptide level features (including various compositional, physicochemical, and structural) were compared. Deep-ABPpred achieved precisions of approximately 97% and 94% on the test dataset and independent dataset, respectively.

Yi et al. [142] proposed a deep LSTM neural network called ACP-DL to improve the ACP predictive power by learning features of the binary profile and *k*-mer sparse matrix of the reduced amino acid alphabets. The output of LSTM layers was fed into a dense layer to obtain the final prediction. ACP-DL achieved the best accuracy of 81.5% and 85.4% compared with SVM, RF, and NB by five-fold CV based on their two benchmark datasets of ACP740 and ACP240, respectively.

BiLSTM is a very popular method in sequence prediction tasks. Noting that antibacterial properties of an AMP may not be relevant to the direction of a peptide sequence, Youmans et al. [143] proposed a BiLSTM model that combined outputs from the forward-processing and backward-processing LSTMs before feeding into a dense classification network. Compared with an RF model that was based on a large number of peptide-level features (45,378 features generated with ProtDCal), the authors found that BiLSTM yielded only insignificant improvement in accuracy. However, because the model required a simpler input with a dimension of only 86, BiLSTM was able to identify relevant features directly from the sequence for identification of AMP, indicating its great potential.

Yu et al. [144] proposed DeepACP, a deep Bi-LSTM learning method, for identifying ACPs. To prove the superiority of the proposed DL method, the authors also built a deep CNN network, as well as a deep CNN and Bi-LSTM fusion network. Finally, the empirical results showed that the proposed DeepACP, which included only Bi-LSTM, was superior to other architectures. Moreover, DeepACP outperformed several existing methods and can be used as an effective tool for the prediction of ACPs.

In addition, fusion methods based on the RNN variants Bi-RNN and GRU were also reported. Hamid et al. [145] identified AMP with word embedding (Word2vec) via a RNN method, which was a two-layer Bi-RNN with GRU, based on their data. They obtained the best result with AUC-PR (area under the curve of precision and recall) of 95.8% compared to other algorithms, including SVM, RF, and logistic regression (with 10-fold CV) based on their dataset.

### 5.4. Hybrid Learning

There have been several research studies on integrating different ML architectures to take advantage of their components. Usually, the different ML architectures have distinct advantages with their particular features and suffer from some disadvantages. The hybrid learning models would theoretically have greater potential in terms of performance improvement and robustness of the application. However, there are also findings that hybrid architectures do not necessarily perform better compared to single methods and meta-learning [146]. Nonetheless, hybrid models keep appearing in the literature, especially using CNN and RNN layers or merging traditional ML methods with DL layers.

#### 5.4.1. Hybrid of CNN and RNN Layers

In the hybrid CNN and RNN, the convolutional layer is used for automatic peptide feature selection to reduce the number of features and, thus, the number of parameters. The LSTM layer is used to identify sequential patterns along the sequence, passing the feature tensor to the next layer at each time step. These aforementioned LSTM gates allow the model to selectively remember or forget the peptide information from the previous step, preventing the gradient vanishing.

Veltri et al. [80] proposed a DL method containing CNN and LSTM layers with the features generated by the Word2Vec embedding method [81]. It showed the best performance among the state-of-the-art methods with an accuracy of 91.0%.

Fu et al. [147] proposed a deep neural network called ACEP that automatically selected and fused the heterogeneous features (PSSM, one-hot, and AAC). ACEP used jointly CNN and LSTM modules and achieved 92.5% accuracy by 10-fold CV based on their dataset.

Fang et al. [148] presented AFPDeep, a deep tandem fusion CNN and LSTM layered network with peptide sequences as input followed by a character embedding layer to generate the sequence representation to identify anti-fungal peptides (AFPs). AFPDeep achieved the best results with AUC-ROC of 95.3% compared to AFPDeep without CNN layers and AFPDeep without LSTM layers on the in-house developed dataset.

Li et al. [65] proposed DeepAVP, a deep fusion CNN and Bi-LSTM architecture with one-hot feature encoding methods as input feature representations to identify anti-viral peptides (AVPs). The one-hot features were input to the CNN branch and the Bi-LSTM branch in parallel, and then the outputs of the two branches were concatenated, followed by a dense layer for prediction. DeepAVP demonstrated the state-of-the-art performance of 92.4% accuracy, which is far better than existing prediction methods for predicting AVPs.

Sharma et al. [149] proposed a DL method called Deep-AFPpred using transfer learning and one-dimensional CNN and Bi-LSTM to identify novel AFPs. Transfer learning was completed by using pre-trained embeddings from seq2vec (PESTV) [150], which trained embeddings from the ELMo language models on millions of protein sequences from UniRef50. Deep-AFPpred achieved the best performance compared to the other state-of-the-art methods with an accuracy of 94.3%.

#### 5.4.2. Hybrid of DL and Attention Mechanism

The attention mechanism aims to simulate human cognitive attention by weighting its input data differently, which results in focusing on the really important parts of the data. The attention mechanism can be divided according to its structure into a single-head attention mechanism [151], a multi-head attention mechanism [84], and a hierarchical attention mechanism [152]. The single-head attention mechanism [151] generates a weighting vector to label the significance of its inputs. The multi-head attention mechanism [84] performs multiple attentions simultaneously and concatenates the outputs together for further processing to learn different emphases. The hierarchical attention mechanism [152] extracts the weight of each input from the *m* input in the first hierarchy, and then treats the *m* input as a unit and computes the weight from the *n* unit, so does the higher hierarchy, to obtain global understanding.

Numerous studies have shown that different parts of a peptide sequence can have different functions, e.g., certain hydrophobic regions can serve as anchors for initial interactions [153]. In this case, attention mechanisms can be applied to detect the important parts that matter for the final prediction output by controlling the weights [154]. For example, the attention mechanism can be combined with RNN to effectively and selectively extract important features of peptide sequences.

Li et al. [117] proposed the AMPlify method, a DL model that included a Bi-LSTM layer and two different attention layers—the multi-head scaled dot-product attention (MHSDPA) layer [84] and hierarchical attention [152] for discovering new AMP. AMPlify achieved 93.7% accuracy, which is the best performance compared to the state-of-the-art methods.

Ma et al. [116] proposed a DL method that combined the prediction results of five different DL models by treating peptide sequences as text and applying NLP methods to form a unified pipeline for identifying AMPs from the human gut microbiome. Among these five DL models, two models had the same architecture of tandem fusion of CNN and LSTM layers based on balanced and unbalanced training datasets, two models with the same architecture of tandem fusion of CNN and attention layers based on balanced and unbalanced training datasets, and finally BERT based on an unbalanced dataset [88]. Experimental results showed the best performance with 99.6% accuracy compared to the state-of-the-art methods.

#### 5.4.3. Hybrid of Traditional ML and DL

DL methods require a large data set to train and are notoriously difficult to tune. However, obtaining experimental data on AMPs is expensive, while a traditional ML with limited data can still achieve a reasonable level of accuracy that cannot be achieved right away with a DL model. Thus, it is tempting to hybridize traditional ML methods and DL methods to generate more comprehensive features that go beyond a single method.

Xiao et al. [96] proposed iAMP-CA2L, a two-layer DL predictor with cellular automata images (CAI) [155] as input features by the method of tandem fusion of CNN, Bi-LSTM, and SVM, to first identify AMPs and then 10 functional classes (ABPs, AVPs, AFPs, anti-biofilm peptides, anti-parasite peptides, anti-HIV peptides, ACPs, chemotactic peptides, anti-MRSA peptides, and anti-endotoxin peptides).

DL methods can be used to extract features of peptides and then learned from traditional ML methods. For example, Sharma et al. [102] proposed AniAMPpred, in which a one-dimensional CNN with Word2vec [81] embedding was used to encode features from peptide sequences and an SVM was used to develop the classifier based on the datasets considering only all available AMPs from the animal kingdom with lengths ranging from 10 to 200 for identifying probable AMPs in the animal genomes.

Singh et al. [64] presented StaBle-ABPpred, a stacked ensemble classifier based on the fusion of Bi-LSTM and attention mechanism for accelerated discovery of AMPs. StaBle-ABPpred is a two-phase architecture that employed word2vec as an embedding layer to extract a feature vector and used peptide sequences as input. In phase 1, after the embedding layer, a Bi-LSTM and an attention layer were connected, and an attention vector was output. In phase 2, with the attention vector as input, the author used three traditional ML methods, namely SVM, LR, and GB, to form an ensemble method for identifying AMPs by majority voting.

### 5.5. The Other DL Approaches for Identifying AMPs

Since the models developed using off-the-shelf DL architectures and transfer learning were both employed by one paper separately, we assigned them to a separate section of the other DL approaches for identifying AMPs.

#### 5.5.1. Off-the-Shelf DL Architectures

In some contributions, DL methods directly based on well-known developed DL architectures were presented. Hussian et al. [156] proposed sAMP-PFPDeep, a deep VGG-16 [157] with three feature encoding methods (features related to the position, frequency, an sum of 12 physicochemical features were considered) to identify short AMPs (peptides with sequence lengths less than 30 residues). sAMP-PFPDeep was compared with RESNET-50 [158] and other state-of-the-art methods, and the results showed that sAMP-PFPDeep performed best with VGG-16 with an accuracy of 84% on an independent data set.

#### 5.5.2. Transfer Learning

Transfer learning aims to reuse learned knowledge from a related task to improve performance on the current task, which is also useful in identifying the AMP activities. Salem et al. [159] proposed a transfer learning method with two pre-trained stages called AMPDeep to identify the hemolytic activity of AMPs that performed best on three different benchmark datasets. In addition to the selected hemolysis data of AMPs for training, they also collected the secretory data for pre-training the transfer learning method. The AMPDeep model was initialized on Prot-BERT-BFD [160], a protein language model trained on approximately 2 billion protein fragments to predict amino acids masked within protein residues.

### 5.6. DL for AMP Regression

In addition to classification tasks, regression models were proposed to predict biological activity assay values of AMPs. Witten et al. [161] proposed a CNN method that combined the classification of AMPs and regression of minimum inhibitory concentrations (MICs). The proposed method achieved a Pearson correlation coefficient (PCC) of 77.0% and an accuracy of 97.0%. Based on the developed method, new AMPs were designed against *Escherichia coli*, *Pseudomonas aeruginosa*, and *Staphylococcus aureus*.

Using multitask learning to supplement the small datasets of cancer-specific peptides, Chen et al. [71] developed xDeep-AcPEP to improve biological activity (EC50, LC50, IC50, and LD50) regression towards six tumor cells, including breast, colon, cervix, lung, skin, and prostate cancers, with an average Pearson’s correlation coefficient of 0.8.

## 6. AMP Design by Optimization

Interest in the development of biologically active peptides can be traced back to the early 1980s when designs were mostly based on sequences of natural peptides. Often, residues that play a role in the structure and/or function of the peptide were selected to be removed or mutated to generate analogous peptides that were then tested for their effects on biological activity. The selection was based on an educated guess or trial and error. These works have contributed significantly to our understanding of the sequence–structure–activity relationships of AMP and demonstrated the power of rational peptide design.

An early exemplary work of AMP design was reported by DeGrado et al. [162], who succeeded in developing an analogous peptide to the bee venom toxin melittin. By optimizing the hydrophobic–hydrophilic balance associated with the amphiphilic α-helical segment in the N-terminal (see Figure 3), the synthetic peptide exhibited the same lytic mechanism as melittin but with increased affinity for the membrane, resulting in higher bioactivity. Although the sequence is not de novo, this pioneering work provides evidence that peptides can be engineered to perform desired functions.

Specific motifs in the sequence of peptides are critical for potent antimicrobial activity. Studies on AMP design, based on the modification of a known AMP sequence as a template, aim to optimize the peptide to achieve greater antimicrobial activity or lower toxicity to human cells. In these studies, peptide sequences were often treated as text, and the sequence composition was altered by mutating amino acids at positions important for antimicrobial activity.

Inspired by natural evolution, the genetic algorithm has been applied to design molecules of targeted properties for a variety of tasks [164,165]. An evolutionary algorithm with ML is a popular solution for improving the antimicrobial activity of template sequences [166,167]. In 2018, Yoshida et al. [168] used evolutionary algorithms and ML to explore sequence spaces to design new AMPs starting with a template peptide. In their work, they used a genetic algorithm with a natural peptide sequence, the in vitro experiment assay as a “fitness” function, and a ML prediction model to provide more efficient prediction to generate the next generation of sequences. Experimental results showed that up to a 160-fold increase in antimicrobial activity was obtained in three rounds of experiments. However, their work relied only on statistical analyses to generate mutations and did not provide useful insights into how to design peptides more effectively using amino acid substitution frequencies.

To gain more transparent knowledge of the AMP design, Boone et al. [169] combined a codon-based genetic algorithm with the rough set theory [170] and a transparent ML approach to improve the understanding of the relationships among specific design solutions in a design space, to design antimicrobial peptides. They started the first generation with known natural AMPs converted to a codon representation to take advantage of reading frames of generating novel AMPs, then mutated a single DNA to generate new peptide sequences, and filtered the newly generated sequences by the high specificity rough set classification method. Experimental results showed that one of the three AMPs selected from the genetic algorithm exhibited antibacterial activity.

However, the above methods, focusing only on amino acid compositions, did not take into account the interactions between amino acids that affected the structures of peptides since the effects of a particular substitution of residues depend on the context in the sequence. Overall, optimization-based AMP design methods have shown some success; however, the optimized peptides are analogs of the original sequence and, thus, much of the sequence space is not explored by this approach.

## 7. De Novo AMP Design

While the earlier work took a natural peptide as a template, Blondelle and Houghten [171] extended the study to artificially designed sequences. They examined a series of sequence analogs consisting only of leucine and lysine residues from the parent sequence Ac-LKLLKKLLKKLKKLLKKL-NH2. On the basis of changes in retention time on reversed-phase high-performance liquid chromatography (RP-HPLC) and antibacterial assays of the analogs with the omitted residues and mutations, they confirmed the importance of an amphipathic α-helical conformation for antimicrobial activity and the need for continuity in the hydrophobic surface of the peptide for hemolysis to occur.

The success of AMP designs relies heavily on prior knowledge and predefined rules discovered from existing AMPs, which are difficult to identify, and are costly and laborious to validate experimentally. On the other hand, design methods based on ML or DL are able to efficiently and effectively explore a large number of sequences by learning various properties of the sequences at the amino acid level and the peptide level.

In the early stage, de novo AMP design often employed DL methods used in natural language processing (NLP) [172]. NLP is an interdisciplinary field of computer science, artificial intelligence, and linguistics. It has two main research directions: natural language understanding and natural language generation. People gradually began to introduce DL to conduct NLP research. The use of DL in NLP has been successfully applied in machine translation [173,174], question–answering systems [175,176], and reading comprehension tasks [177]. RNN is one of the most widely used methods for NLP [94,178], models such as GRU [179] and LSTM [141] have sparked wave after wave of upsurges. In recent years, pre-trained language representation models have been developed that initially perform large-scale unsupervised or self-supervised learning pre-training before downstream tasks such as ELMo [180], GPT [181], BERT [182], and so on. These models are proven to be far more powerful than traditional language models in most NLP tasks. The modular nature of peptides, with each amino acid acting as a word and together forming a sentence, has inspired researchers to draw linguistic models for AMP understanding and generation with the patterns as grammatical rules and the amino acids as vocabulary [183].

RNN is a powerful method for sequential data learning. It uses memory cells to remember the information of each input by processing the entire sequence but only one input (e.g., a word) at a time. Some successes have been reported with RNN on AMP de novo design with good performance [184,185]. The major limitation of RNN is the gradient from vanishing and exploding problems, which is overcome by LSTM.

This improved version of the memory cell uses gate mechanisms, such as input gates, output gates, and forget gates to ensure that information is processed properly and uses a backward loop to ensure that the error signal in the form of a gradient is not lost after processing a long sequence. For example, Müller et al. [68] trained RNN with LSTM units for combinatorial de novo AMP design. The network focused on linear cationic peptides forming amphipathic helices, which are considered the most relevant properties for antimicrobial activity. The network was trained to predict the next amino acid for each position in the input. De novo sequence generation was performed by predicting the amino acids until an empty character or a maximum length of 48 residues was reached. Of the 2000 sequences generated, 85% of the generated sequences were predicted to be active AMPs by the CAMP prediction tool [186].

Another DL architecture, variational autoencoders (VAEs), are also popular in generating new chemical spaces [187,188,189]. A VAE consists of an encoder, which converts the molecule into a latent vector representation, and a decoder, in which the latent representation attempts to recreate the input molecule. VAEs follow an encode–decode model, which supports the random generation of latent variables and improves the generalization ability of the network.

Dean et al. [190] demonstrated the use of a VAE for de novo AMP design, the model was trained on thousands of known and scrambled AMP sequences from APD3 and they experimentally verified generated peptides to be active. Later on, Das et al. [191] proposed a computational method employing VAE to leverage the guidance from classifiers trained on an informative latent space and then checked the generated molecules with the DL classifiers. Finally, they identified, synthesized, and experimentally tested 20 generated sequences, two of which displayed high potency against diverse Gram-positive and Gram-negative pathogens.

Generative adversarial neural (GAN) networks [192,193] have become very popular architectures for generating highly realistic content in recent years. A GAN has two components, a generator and a discriminator, which compete against each other during training. The generator produces artificial data and the discriminator attempts to distinguish it from real data. The model is trained until the discriminator is unable to distinguish the artificial data from the real data.

GAN uses the competitive path and no longer requires an assumed data distribution. However, since it has no loss function, there is the problem of collapse, which makes it difficult to determine whether progress is being made [194].

Tucs et al. [195] proposed PepGan, which uses a GAN to find the balance between covering active peptides and avoiding non-active peptides. They synthesized the six best peptides, and one peptide has a significantly lower MIC against *E. coli* than ampicillin.

Oort et al. proposed AMPGAN v2 [69] using a bidirectional conditional GAN (BiCGAN) to learn data-driven priors and control generation using conditioning variables. They then validated the generated AMP candidates using CAMPR3, and a high percentage (89%) of the generated samples were predicted to be anti-microbially effective.

Table 2 summarizes some representative works on the de novo design of AMPs. We can see that the generated peptides are usually short, not longer than 30 amino acids with different target species, while most of them have strong antimicrobial activity, indicating the good performance of de novo design methods.

## 8. Limitations and Challenges

### 8.1. Data Insufficiency

The use of DL in the discovery and design of AMPs has led to labor and time savings. These DL models are mostly based on supervised learning and require large datasets of validated AMPs to train. Compared to problems in other domains, such as computer vision and natural language processing, where DL methods are used extensively, the amount of AMP data available (millions of data samples versus a few tens of thousands of AMPs) is tiny. Therefore, how to overcome the data limitation problem is of fundamental importance.

With the advances in semi-supervised and unsupervised learning, the research community has begun to utilize the freely available high-throughput sequencing data to train general DL models to learn amino acid sequences. The massive amount of data does not provide direct information about the antimicrobial activities of peptides, but the language models or the embeddings [89,137,138] may help to represent and extract the inherent biological properties between amino acids that naturally exist in the sequences of organisms. It remains to be seen whether these latest DL techniques can be successful in discovering truly novel, potent, and therapeutically effective AMPs.

In the field of image classification, data augmentation is a solution to the problem of limited data. It consists of a set of techniques to generate new examples based on existing images. The enlarged sizes and quality of image data for training lead to improved model performances [197]. However, data augmentation techniques for predictive problems involving biological sequences have not been extensively explored. Some improvements have been reported based on simple augmentation approaches, such as feature perturbation [198], random substitutions, and insertions [199]. Sample sequence generation based on GAN [200] has also shown promise, but requires training of models specific to each application and, thus, has limited generalizability [199]. Innovative methods for biological sequence data augmentation are therefore needed.

We must also point out that a large amount of data does not necessarily guarantee significant performance improvements. Datasets are highly susceptible to noise, missing values, and data inconsistencies because a large dataset is usually compiled from multiple heterogeneous data sources. Take DBAASP [7] as an example, which is a popular database for antimicrobial/cytotoxic activity and the structures of peptides; for the same peptide, there may be multiple measurements from multiple laboratories with different experimental conditions. Researchers must carefully verify the relevance of the data through experimentation to ensure the veracity of the data. In this case, the preprocessing of data is a necessary step before feeding it into DL models. Data cleaning methods that deal with missing values and noise can be useful to improve the quality of a dataset, including accuracy, consistency, and so on.

### 8.2. Limited Modeling beyond Binary Classification of Linear AMPs

So far, most AMP prediction methods are binary classifications, i.e., only AMPs and non-AMPs are predicted. Since an AMP can have markedly different activities, such as antibacterial, antifungal, antiviral, and anti-parasitic [201], it is valuable to make predictions about the type of biological activity a peptide targets. This multi-class or multi-label prediction problem is more demanding on datasets where imbalanced and missing data attributes impose major challenges for training DL models.

Besides recognition, AMP classifiers are often used to prioritize candidate sequences for experiments. However, the classification probability of a sequence predicted by the classifier and its actual antimicrobial activity of being strong, moderate, or weak, do not necessarily correlate, resulting in low enrichment in the ranked list. To correlate the predicted value to the bioactivity assay means training a regression model, a few attempts of such a development were reported [161,202,203] but the predictive performance is far from satisfactory. The challenges are limited; there are noisy experimental bioactivity data and a lack of approaches to unify different measurements from experiments.

It is worth noting that the function of AMP can be further improved by chemical modifications [204], metal complexation [205], or by building various micro- or nanostructures [206]; therefore, in silico methods to support the design of AMPs with these additional components and biological processes are in need.

### 8.3. Limited Attempt in Drug-Likeness Prediction of AMPs

Improving the pharmacokinetic properties is critical for successful therapeutic application of AMPs. In fact, the major limitations of AMPs to become real drugs are their short half-life, cell toxicity, and unexpected side effects [207].

AMPs, similar to other peptides, are susceptible to enzymatic degradation, resulting in poor bioavailability. Experimental strategies to improve the stability of a peptide include cyclization, incorporation of non-canonical amino acids, and terminal modifications. However, in silico prediction of peptide stability would be invaluable in providing an initial assessment of degradation potential as well as the location of the cleavage site, thus providing guidelines for sequence optimization prior to chemical synthesis. To date, only a few ML methods have been developed to predict the half-life of peptides. For example, PlifePred used PaDEL descriptors and generated classical ML (SVM) models that achieved correlations in the range of 0.65 to 0.75 for predicting the peptide half-life in mammalian blood [208]. Their earlier work provided another classical ML model that predicted peptide half-life in gut-like environments [209]. For short peptides, tools for drug-likeness assessment, such as SwissADME [210], ProTox-II [211], are available although these tools were originally developed for small molecular compounds. SwissADME, for example, can test peptides with up to six amino acids.

Another important challenge in AMP research is to consider toxicity in addition to antimicrobial activity. An example is vancomycin, which is FDA-approved but can cause kidney damage in some patients or at high doses [207]. There are studies in AMP design that predict the toxicity of AMPs to humans while maintaining their efficacy. In 2013, Gupta et al. developed ToxinPred [212], an online tool to predict the toxicity of peptides based on SVM. More recently, in 2020, Taho applied transfer learning to predict the host toxicity of antimicrobial peptides [213].

As can be observed, the performance of existing ML or DL methods for predicting the stability and toxicity of peptides is far from satisfactory. The problems may arise from the lack of large experimental data sets and non-standardized measurement reporting procedures, which result in confusing or noisy data. In addition, as with AMP prediction, the classification results do not provide biological insights that can guide the next stage of optimization but are important to improve the drug properties of the peptide for successful therapeutic application.

### 8.4. DL Model Optimization and Reproducibility

Apart from the need for suitable training data, there are still several limitations in the selection of feature encoding methods and the training process of DL models, such as parameter settings [214] and reproducibility [215]. There are many different feature encodings of peptide sequences. Finding a set of appropriate feature encoding methods for different DL models and different prediction tasks is a big challenge. Although the approaches of DL have proven to be powerful and promising for the discovery and design of AMPs, there are a number of parameters during the training process, and these parameters need to be well-designed and adjusted depending on the DL models.

In recent years, the irreproducibility of DL models has raised concerns about the reliability of many academic works [216,217,218]. To address this issue, the source code and data, including training and test data, should be made openly available and well documented in the interest of reuse.

### 8.5. Explainable Artificial Intelligence

Rational peptide drug designs require an understanding of the functions of molecules and the relationship between functions and their primary and three-dimensional structures. AMP prediction models help distinguish between active and inactive candidates by classification or predict activity by regression, but often do not provide explanations for the prediction results. These black boxes add only limited contributions to our understanding of biology, and worse, they make the prediction results less trustworthy for experimenters.

Explainable artificial intelligence (XAI) [219] is an emerging subfield of AI. XAI-generated explanations can be categorized into global and local explanations. The former summarizes the relevance of input features in the model, while the latter is based on individual predictions [220]. Since XAI depends on the underlying model, there is no “one-fits-all” XAI approach.

For predicting AMPs, it is challenging to provide explanations based on input features. The problem is the lack of a simple input representation for sequence data, such as the corpus in natural language processing. The representation or encoding itself is a complex combination of different information, including the compositional, physicochemical, structural, and evolutionary properties of amino acids. While the choice of representation or encoding is critical to the performance of the model, it becomes a limiting factor for generating understandable and helpful interpretations with XAI.

## 9. Conclusions

In this work, we provided an overview of state-of-the-art DL methods for AMP discovery and design. To this end, we first introduced AMPs, including retrospecting their discovery histories, properties, structural classifications, action mechanisms, and therapeutic and industrial applications. Subsequently, the computational workflow of AMP discovery and design based on traditional ML and DL is presented. Following the workflow, feature encoding methods, a summary of traditional ML methods, and DL methods for discovering AMPs, as well as an AMP design based on template sequences and the de novo AMP design were reviewed. Finally, the limitations and challenges for AMP discovery and design, such as insufficient data and explainable artificial intelligence, were discussed.

## Figures and Tables

**Figure 1 antibiotics-11-01451-f001:**
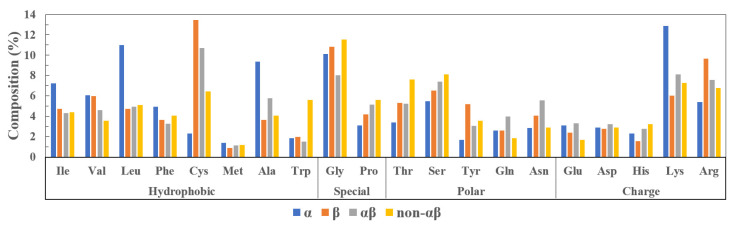
Comparison of the amino acid compositions of four AMP classes (α, β, αβ, and non-αβ) based on the 721 records of structurally-annotated natural peptides in the APD3 database.

**Figure 2 antibiotics-11-01451-f002:**
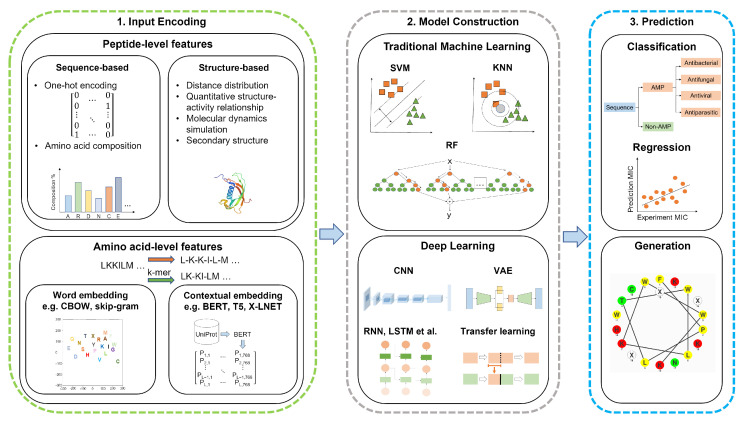
A general ML workflow of AMP discovery and design, including a summary of the major techniques in each stage of the workflow.

**Figure 3 antibiotics-11-01451-f003:**
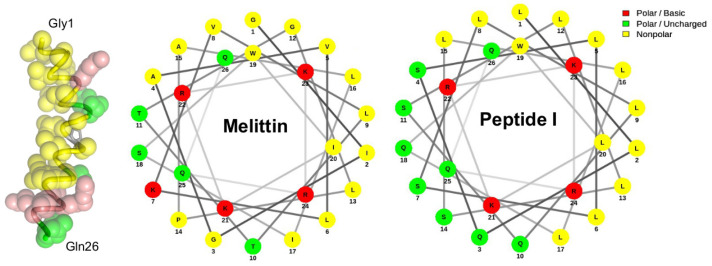
Comparison of the amino acid composition of four AMP classes (α, β, αβ, and non-αβ) based on the 721 records of structurally-annotated natural peptides in the APD3 database. proposed that the amphiphilic helix (residues 2–13) is the main driving force for membrane binding. Encouragingly, peptide 1 is engineered with increased amphiphilicity in this segment, resulting in enhanced binding and lytic activity, as confirmed in their experiments. Images were created using PyMOL and NetWheels [163].

**Table 1 antibiotics-11-01451-t001:** Summary of the average length and net charge of the four classes of AMPs in the APD3 database.

	α	β	αβ	Non-αβ	Unknown
**Total no. of peptides**	494	89	120	22	2710
**Average length**	29.66	35.08	58.92	26.82	--
**Average net charge**	+3.63	+3.65	+5.37	+2.55	--

**Table 2 antibiotics-11-01451-t002:** Studies of de novo design with successful novel sequences and experimental assay values.

Technique	De Novo Sequence	Length	Target Species	Activity MIC (μg/mL)	Reference
Manually designed	Ac-LKLLKKLLKKLKKLLKKL-NH_2_	18	*S. areus* *E. coli* *P. aeruginosa*	64 64 128	[171]
Ensemble learning, ANN	ALFGILKKAFGKILTIFAGLPGVV	MCF7 A549	9.8 (EC_50_) 8.6	24	[196]
	GLGDFIKAIAKHLGPLIGILPSKLKVAA	28	MCF7 A549	4.5 11.3	
	FLGPTIGKIAKFILKHIVGLGDAALV	26	MCF7 A549	2.6 10.7	
	GLFAILKKLVNLVG	15	MCF7 A549	2.3 4.6	
	GLFKIISKLAKKA	13	MCF7 A549	27.7 36.3	
VAE	KKIKRFLRKIG	11	*E. coli* *A. baumannii* *S. aureus*	11 36 0.4	[190]
	KLFRIIKRIFKG	12	*E. coli* *A. baumannii* *S. aureus*	0.2 0.8 >400	
VAE, LSTM	YLRLIRYMAKMI-CONH_2_	12	*S. aureus* *E. coli* *P. aeruginosa* *A. baumannii* MDR *K. pneumoniae* polyR *K. pneumoniae*	7.8 31.25 125 15.6 31.25 31	[191]
	FPLTWLKWWKWKK-CONH_2_	13	*S. aureus* *E. coli* *P. aeruginosa* *A. baumannii* MDR *K. pneumoniae* polyR *K. pneumoniae*	15.6 31.25 62.5 31.25 15.6 16	
GAN	ILPLLKKFGKKFGKKVWKAL IKALLALPKLAKKIAKKFLK GLRSSVKTLLRGLLGIIKKF GLKKLFSKIKIIGSALKNLA FLPAFKNVISKILKALKKKV FLGPIIKTVRAVLCAIKKL	20 20 20 20 20 20	*E. coli* *E. coli* *E. coli* *E. coli* *E. coli* *E. coli*	25 50 >100 2.1 12.5 25	[195]

## Data Availability

Not applicable.

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
