# Peer review of "Recent Progress in the Discovery and Design of Antimicrobial Peptides Using Traditional Machine Learning and Deep Learning"

_antibiotics, 2022, doi:10.3390/antibiotics11101451_

Round 1

Reviewer 1 Report

The article, “Recent Progress in the Discovery and Design of Antimicrobial Peptides using Traditional Machine Learning and Deep Learning”, surveys the latest AMP prediction methods with an emphasis on deep learning approaches.  Overall, I find this to be an excellent review; it is informative and organized very well. While review articles on AMPs already exist (the authors in fact point out several in their introduction) the scope of this article covers ML and DL approaches to AMP prediction methods that aren’t as well reviewed in the literature. The article very clearly lays out the research that led to our current state of understanding. This would serve as a helpful resource for any researcher interested in learning about contemporary methods of AMP prediction, The authors provide a narrative that is informative and interesting. In particular, I found Figure 2 to be a very elegant representation of the main objectives of this review. 

One suggestion I have is to note the limitations of AMPs in the introductory material. While the introduction comprehensively reviews the bactericidal and bacteriostatic properties of AMPS, their benefits as alternatives to conventional antibiotics, and interesting historical perspectives, it does not substantially address the limitations of AMP application. There are most definitely reasons that AMPs do not get used as widely as conventional antibiotics. This may be due to issues of availability or cost, but perhaps just noting a concrete limitation of AMP efficacy in certain scenarios would be enough. As an example, some bacteria have developed resistance to AMPS; gram negative bacteria have a mechanism to detect AMPS and neutralize their activity via proteases (see article “Resistance to antimicrobial peptides in Gram-negative bacteria” FEMS Microbiology Letters, Volume 330, Issue 2, May 2012, Pages 81–89). Pointing out the limitations of AMPs in the introduction does not detract from their utility but will lend a broader perspective on their use in various applications.

Besides this I have some relatively minor line-by-line suggestions:

2          Do you mean the abuse of antibiotics (instead of antibodies)?

27        Please provide a citation for AMPs as anticancer agents

76/78  Use 1960s/1980s instead of 60s/80s for consistency

116-117           For clarity, note the charges as +3.6, +5.4, +2.55 (its implied by the previous statement that most peptides are cationic, but adding the + avoids any ambiguity). Also, check the significant figures for these values; 2.55 has 3 sig figs but the other reported values only have two. Just verify whether 3.6 and 5.4 should be reported instead as 3.60 and 5.40. 

271 italicize de novo

Author Response

The responses can be found in the attachment.

Reviewer 2 Report

This manuscript thoroughly reviews the application of machine learning and deep learning in the discovery of antimicrobial peptides. The manuscript is well written, and the quality of the presentation is excellent.

The main issue with the therapeutic usage of AMPs is their poor pharmacokinetics (https://www.mdpi.com/2079-6382/9/1/24). In my opinion, authors should explain if machine-learning techniques could also aid the design and discovery of AMPs with improved drug-likeness.

Some minor found in the text:

Abstract – line 2 – antibodies should probably be replaced with antibiotics

Line 22 – insect should be replaced with insects

Line 27 – “suggesting them a potential” should be replaced with …them as a potential…

Author Response

(The authors gave the same response as above.)

Reviewer 3 Report

The review entitled "Recent Progress in the Discovery and Design of Antimicrobial Peptides using Traditional Machine Learning and Deep Learning" has an important focus and deserves consideration but as such it is not so different from hundreds of reviews already present in the literature. I think that the authors should try to give a different cut to the manuscript avoiding the traditional structural classification or reducing it to minimum terms. Thus I advise reducing the first part of the review to streamline reading. 

As a minor point, I advise providing insights also in metallo-based AMPs as reported in Antibiotics (Basel). 2020 Jun 18;9(6):337. doi: 10.3390/antibiotics9060337 and self-assembly peptides as AMPs Int J Mol Sci. 2021 Nov 23;22(23):12662. doi: 10.3390/ijms222312662.

Author Response

(The authors gave the same response as above.)
